# Laryngeal Mid-Cord Erythroleukoplakias: How to Modulate the Transoral CO_2_ Laser Excisional Biopsy

**DOI:** 10.3390/cancers12082165

**Published:** 2020-08-04

**Authors:** Francesco Mora, Filippo Carta, Francesco Missale, Andrea Laborai, Giampiero Parrinello, Cesare Piazza, Roberto Puxeddu, Giorgio Peretti

**Affiliations:** 1IRCCS Ospedale Policlinico San Martino, 16132 Genova, Italy; francesco.mora@unige.it (F.M.); giampiero.parrinello@gmail.com (G.P.); giorgioperetti18@gmail.com (G.P.); 2Department of Surgical Sciences and Integrated Diagnostics (DISC), Unit of Otorhinolaryngology, University of Genoa, 16132 Genoa, Italy; labo88@fastwebnet.it; 3Department of Otorhinolaryngology—Head and Neck Surgery, University of Cagliari, 09124 Cagliari, Italy; filippocarta@unica.it (F.C.); puxeddu@unica.it (R.P.); 4Department of Molecular and Translational Medicine, University of Brescia, 25121 Brescia, Italy; 5Department of Otorhinolaryngology, Guglielmo da Saliceto Hospital, 29121 Piacenza, Italy; 6Department of Otorhinolaryngology, Maxillofacial and Thyroid Surgery, Fondazione IRCCS, National Cancer Institute of Milan, 20133 Milan, Italy; ceceplaza@libero.it; 7Department of Oncology and Oncohematology, University of Milan, 20122 Milan, Italy

**Keywords:** laryngeal neoplasms, narrow band imaging, Storz professional image enhancement system, stroboscopy, saline infusion, lasers, gas, cordectomy, transoral, squamous intraepithelial lesions

## Abstract

Background: The endoscopic appearance of glottic erythroleukoplakias is non-predictive of their histopathology, potentially ranging from keratosis to invasive squamous cell carcinoma (SCC). The aim of this study was to assess a comprehensive workup for the one-step diagnosis and treatment of mid-cord erythroleukoplakias, using CO_2_ laser excisional biopsy. Methods: We evaluated 147 untreated patients affected by 155 mid-cord erythroleukoplakias submitted to excisional biopsy by subepithelial (Type I) or subligamental cordectomy (Type II), across two academic institutions. Patients were evaluated by preoperative videolaryngostroboscopy, pre- and intraoperative videoendoscopy with biologic endoscopy (narrow band imaging, NBI, or the Storz professional image enhancement system, SPIES), either with or without intraoperative saline infusion into the Reinke’s space. Adequacy of treatment was the primary outcome. Results: The histopathologic diagnosis was keratosis in 26 (17%) cases, squamous intraepithelial neoplasia (SIN1-2) in 47 (30%), carcinoma in situ in 21 (14%), and SCC in 61 (39%) patients. The adequacy of treatment across the entire cohort was 89%. The intraoperative saline infusion procedure, facing not clearly suspicious lesions, raised the adequacy of treatment from 60% to 90% (*p* = 0.006). Conclusions: Excisional biopsy by Type I–II cordectomies, after a comprehensive diagnostic workup, should be accepted as an adequate and cost-effective treatment of unilateral mid-cord glottic erythroleukoplakias.

## 1. Introduction

The diagnosis and treatment of premalignant and early malignant glottic lesions (arising, in most cases, as leuko- or erythroleukoplakias) are still challenging due to the need for correct one-step diagnosis and treatment by excisional biopsy, which aims for the best compromise between oncological radicality and satisfactory vocal outcomes. Moreover, to date, there is no single optical biopsy technique that is able to preoperatively elucidate the nature and biological behavior of a given mucosal lesion with an accuracy comparable to that of definitive histopathology. The latter, therefore, remains the gold standard for defining the real nature of a given epithelial lesion. The only certainty, to date, is represented by the fact that the pure endoscopic appearance of glottic erythroleukoplakias cannot be considered as pathognomonic of a definite final histology, potentially ranging from keratosis without atypia to different degrees of squamous intraepithelial neoplasia (SIN), then up to microinvasive or invasive squamous cell carcinoma (SCC) [1].

To further complicate the situation, multiple biopsies (with or without frozen section analysis) are frequently inadequate for obtaining a precise diagnosis and subsequently guiding the ensuing treatment of glottic leuko- or erythroleukoplakias, since they are not representative of the entire lesion and less suitable for further biomarker analysis, which is sometimes needed to provide an accurate differential diagnosis [2]. Furthermore, multiple incisional biopsies may cause undue damage and fibrosis to the multi-layered and delicate microanatomy of the vocal cord [3], forcing the surgeon to subsequently perform a wider and deeper resection, in terms of transmuscular cordectomy (Type III according to the European Laryngological Society (ELS) classification [4]), in order to excise the tumor and surrounding cicatricial tissues. In such a scarred vocal cord scenario, in fact, all the sophisticated diagnostic techniques herein described would become useless or, at least, greatly hampered [5,6].

Therefore, the ideal management of such lesions should be through excisional biopsy by transoral complete excision within ultra-narrow, free surgical margins [7,8], with subepithelial (Type I) or subligamental (Type II) cordectomies, according to the ELS classification [4]. The choice between these two types of cordectomies, aimed at balancing the radicality and quality of the voice, should be modulated based on clinical and endoscopic data coming from a sophisticated diagnostic workup such as that already described by the authors in 2003 [9].

In fact, many technical innovations and endoscopic tools have been recently developed to better evaluate such lesions in the pre- and intraoperative settings, including white light (WL) videoendoscopy and different kinds of biologic endoscopy, such as narrow band imaging (NBI) and the Storz professional image enhancement system (SPIES) [6,10,11]. These techniques, together with videolaryngostroboscopy (VLS) and saline infusion (SI) into the Reinke’s space, help in obtaining detailed information about the tissue characteristics, vascular network, and superficial versus deep involvement of the different layers of the lamina propria by the lesion itself, thus allows for a tentative diagnosis (at least in terms of intraepithelial or more invasive disease) and adequate resection using a one-step excisional biopsy approach [9,12,13].

The aim of this study was to assess the overall adequacy of a refined diagnostic workup for obtaining, in one-step, both the correct diagnosis and adequate treatment of mid-cord glottic erythroleukoplakias.

The pre- and intraoperative workup included preoperative flexible videoendoscopy and VLS, as well as intraoperative endoscopy with WL and biologic endoscopy (NBI or SPIES) assessment. According to the gathered information, patients were non-randomly divided into three groups of treatment. Group A was composed of patients harboring highly suspicious lesions, who were straightforwardly treated by a subligamental cordectomy (Type II). Groups B and C were composed of patients presenting not clearly suspicious lesions, in which the data emerging from the diagnostic workup did not suggest SCC. In Group B, upfront excisional biopsy by a subepithelial (Type I) cordectomy was performed, whereas in Group C, further investigation with intraoperative SI guided the final choice of performing a subepithelial (Type I), or subligamental (Type II) cordectomy [13].

The main outcome was the adequacy of the treatment, while the second one was represented by the investigation of the potential value of SI testing in not clearly suspicious lesions.

## 2. Materials and Methods 

### 2.1. Study Design

A retrospective study was carried out, enrolling 147 previously untreated patients (male/female ratio, 5/1; mean age, 66 years; age range, 25–88 years), 139 (94.5%) of whom were affected by unilateral mid-cord erythroleukoplakias, and 8 (5.5%), by bilateral mid-cord lesions without involvement of the anterior commissure, who underwent a total of 155 excisional biopsies by transoral CO_2_ laser microsurgery (CO_2_ TOLMS according to the ELS nomenclature [14]). The procedures were performed between January 2012 and December 2016 at the Departments of Otorhinolaryngology—Head and Neck Surgery at the Universities of Genoa (N = 111) and Cagliari (N = 44), Italy. The extent of the resection was graded according to the ELS classification as subepithelial (Type I) and subligamental (Type II) cordectomies [4,5].

The exclusion criteria encompassed (1) patients previously biopsied and/or treated elsewhere; (2) affected by lesions involving the anterior commissure; (3) extending toward the supra- and/or subglottis; (4) with impairment of vocal cord mobility; (5) with suspicious pre- or intraoperative vocal muscle involvement needing deeper resections, such as transmuscular (Type III) or total (Type IV) cordectomies [4].

### 2.2. Diagnostic Workup

All patients were preoperatively evaluated by WL, biologic endoscopy (NBI in 111 cases and SPIES in 44), and VLS (performed in the office with a 70° rigid endoscope and a Kay Digital Strobe 9200 (Kay Elemetrics Co., Pine Brook, NJ, USA) or a flexible videoendoscope coupled with an OLYMPUS CLL-S1 Strobe LED light source (Olympus, Tokyo, Japan)). Just before surgery, in the operating room, with the patient under general anesthesia, intraoperative WL and NBI or SPIES endoscopy, by angled (0° and 70°) telescopes, was also systematically performed. According to the pre- and intraoperative information gained, patients were non-randomly divided into 3 groups of treatment. Group A was composed of patients who, according to the above described workup, were suspected of harboring lesions highly suspicious for SCC transgression in the epithelial basal membrane due to an absence of a mucosal wave at VLS and/or the presence of atypical vascular patterns at the biologic endoscopy evaluation, according to the ELS classification [15]. These subjects were straightforwardly treated with a subligamental cordectomy (Type II). 

Groups B and C were composed of patients presenting not clearly suspicious lesions, for which the data emerging from the diagnostic workup were not univocal or indicated a possible intraepithelial lesion, meaning the presence of a normal mucosal wave at VLS and absence of intrapapillary capillary loops at biologic endoscopy evaluation. In Group B, upfront excisional biopsy by a Type I subepithelial cordectomy, sparing the vocal ligament, was performed based on the presence of a normal mucosal wave at VLS. In Group C, further investigations to better define the possible involvement of the vocal ligament were intraoperatively added; thus, SI was performed in the Reinke’s space using a dedicated angled needle (Endocraft LLC, Boston, MA, USA) [13]. In this latter group, subepithelial cordectomy (Type I) was carried out when complete mucosal “ballooning” was achieved by SI, while the extension to the subligamental plane (Type II) was reserved for all cases with an absent or incomplete “hydrodissection” after SI. Two exemplificative cases are presented in Figure 1.

### 2.3. Treatments

The surgical procedures encompassed CO_2_ TOLMS subepithelial (Type I) and subligamental (Type II) cordectomies, as defined according to the ELS nomenclature [14]. Patients were intubated with a laser-safe endotracheal tube 5.0–6.0 mm in internal diameter (Shiley™ Laser Oral Endotracheal Tubes, Medtronic Xomed, Jacksonville, FL, USA) under general anesthesia. The laryngeal exposure was achieved with large-bore laryngoscopes (Microfrance 121; Microfrance, Paris, France) or small-bore ones for unfavorable exposures (Dedo-Ossof; Pilling, Philadelphia, PA, USA). The Boston University suspension system (Pilling; Philadelphia, PA, USA) was employed. A UltraPulse^®^ DUO CO_2_ Laser coupled to a Digital AcuBlade scanning micromanipulator (Lumenis, Yokneam, Israel), set on continuous ultrapulse mode (line shape 0.9–1.2 mm) at 2.5–3.5 watts of delivered power and 400 mm of focal length, was used as the cutting device.

The subepithelial (Type I) cordectomy was performed resecting the vocal fold epithelium, with the deep plane passing through the superficial layer of the lamina propria and thus sparing the deeper layers and vocal ligament [14]. Subligamental (Type II) cordectomy included the resection of the epithelium, Reinke’s space, and vocal ligament, with the deep plane of resection being the spared vocal muscle [14]. To expose the entire vocal fold, partial resection of the ventricular band may be necessary. For both types of cordectomy, the specimen is en bloc resected and oriented for proper histopathologic assessment of the surgical margins.

### 2.4. Outcome Evaluation and Statistical Analysis

The adequacy of treatment, in terms of the type of cordectomy chosen to minimize damage to the multilayered vocal cord anatomy—according to the findings of the above-mentioned diagnostic workup—was calculated in relation to the final histopathological report (Table 1). In this perspective, Type I cordectomy was judged as adequate when performed for lesions limited to the superficial layer of the lamina propria (keratosis without atypia or squamous intraepithelial neoplasia (SIN)), while Type II cordectomy was considered adequate for the treatment of both intraepithelial neoplasia or microinvasive/invasive SCC and should be considered an overtreatment for keratosis without atypia (Table 1) [9,16,17]. Comparisons were performed with Fisher’s exact or Chi-square tests, as appropriate. GraphPad Prism Version 6.0 (San Diego, CA, USA) and R (version 3.6.3) were used for statistical analysis. For all tests, a two-tailed *p* value < 0.05 was considered significant.

## 3. Results

### 3.1. Patient Cohort

One-hundred and forty-seven previously untreated patients were enrolled: 139 (94.5%) affected by unilateral mid-cord erythroleukoplakias and 8 (5.5%), by bilateral mid-cord lesions, without involvement of the anterior commissure. They underwent a total of 155 excisional biopsies by transoral CO_2_ laser microsurgery, as described in Table 2.

The histopathology reports show keratosis without atypia in 26 (17%) cases, SIN 1-2 in 47 (30%), carcinoma in situ (CIS) in 21 (14%), and microinvasive or invasive SCC in 61 (39%) (Table 2).

### 3.2. Univariable Adequacy of Treatment Analysis

Histology was significantly associated with endoscopic findings (*p* < 0.001) (Table 3). The overall adequacy of treatment, as defined in Table 1, was 89% for the entire cohort, and 99%, 60%, and 90% for Groups A, B, and C, respectively (Table 2 and Table 4). Among the non-suspicious lesions of Group C, SI changed the surgical planning in 11 (18%) cases, raising the adequacy of treatment from 60% of Group B (in which SI was never performed) to 90% (*p* = 0.006). In Group B, in fact, 10 (40%) patients affected by microinvasive SCC were erroneously undertreated by Type I cordectomy (Table 2). In Group C, the incomplete mucoligamentous hydrodissection came after SI highlighted a microinvasive SCC in 36.4% of cases, compared to only 8% of false negative cases with complete hydrodissection (Table 2). SI allowed more adequate treatment for four patients affected by SCC and, therefore, treated by Type II cordectomy (Table 4).

### 3.3. Multivariable Adequacy of Treatment Analysis

Investigating by a multivariable logistic regression model the treatment Group, the Center at which surgery was performed, and the cordectomy applied as possible confounders, both Group A (OR 47.29, CI95% (3.66–637.88), *p* = 0.002) and Group C (OR 9.58, CI95% (2.63–42.52), *p* = 0.001) were confirmed to be associated with a significantly higher probability of adequacy of treatment, compared to Group B. No significant difference was observed for the Cagliari center compared to the Genoa one (OR 0.3,2 CI95% (0.09–1.00), *p* = 0.052), or for the Type II compared to the Type I cordectomy (OR 0.46, CI95% (0.07–3.82), *p* = 0.42), as shown in Figure 2a,b.

### 3.4. Margin Control

Considering the 82 patients affected by SCC, for 7 (8.5%), the deep margin of resection was involved at the final histopathologic evaluation. These cases were managed with adjuvant radiotherapy in three cases, re-resection in two, and a wait-and-see policy in two. We observed a significantly higher rate of deep positive margins (*p* = 0.03) in patients treated by Type I cordectomy (24%) than in those treated by Type II (5%). Moreover, all patients with deep positive margins after Type I cordectomy belonged to Group B (and were therefore not intraoperatively assessed by SI).

## 4. Discussion

A general consensus in the literature has been achieved on the need to surgically cure mid-cord erythroleukoplakias by excisional biopsy [9,16,17,18], as the risk of the malignant progression of such lesions ranges from 1.6% for low-grade to 12.5% for high-grade dysplasia, and up to 40% for CIS [1]. By contrast, multiple random biopsies or mucosal stripping for “diagnostic” purposes alone, without a proper orientation of the surgical specimen, do not provide reliable information on the nature of the entire lesion. In fact, such inadequate procedures might lead to incorrect diagnosis, and invariably create scar tissue and fibrotic changes in the vocal fold [9,19]. On the other hand, even if the endoscopic treatment represents the best choice for dysplasia and CIS, according to the NCCN guidelines since 2014 [8], how best to modulate the excisional biopsy within peripheral and, more importantly, deep free margins is still a matter of debate. In light of this, the routine application of the aforementioned pre- and intraoperative diagnostic workup is extremely useful in defining the superficial extension of the lesion and its invasion of the lamina propria, and in predicting, in most cases, the final histology, with a consequent improvement in choosing the ideal treatment. 

The most important adjunct in the present workup, compared to that already described by us in 2003 [9], is represented by the routine use of biologic endoscopy to discern premalignant/malignant lesions from innocuous ones. In this sense, many different biologic endoscopy techniques have been described in the literature [6], even though their use frequently relies on purely subjective preferences. For example, we applied NBI and SPIES at two different institutions that share a common treatment policy, obtaining highly comparable diagnostic accuracy. Indeed, even though they are based on slightly different engineering details, both these techniques aim to enhance the vascular network of mucosal lesions and thus differentiate them according to their biologic potential. As already demonstrated by Stanikova et al., NBI and SPIES appear to provide comparable outcomes with excellent diagnostic agreement between them [11]. Furthermore, coupling biologic endoscopy with contact endoscopes is a promising and inter-rater-reliable technique to further improve the diagnostic accuracy of intraoperative endoscopic evaluation [20,21].

Although biologic endoscopy enhances the abnormalities of the epithelium, improving the detection of a precancerous or cancerous lesion in the whole upper aero-digestive tract, it has some intrinsic limits in defining the deep extension of a given lesion. At the glottic site, combining VLS can predict the depth of infiltration on the basis of impairment or absence of the mucosal wave [15,22]. Moreover, the synergy between biologic endoscopy and SI intraoperatively helps the surgeon, especially when the findings of the VLS and biologic endoscopy are equivocal [23]. In fact, when invasion of the vocal ligament is not preoperatively suspected by VLS and biologic endoscopy is inconclusive on the most probable pathologic diagnosis (especially faced with keratotic lesions), thereby masking the vascular features of the mucosal and submucosal layers by an “umbrella effect” [15], the use of intraoperative SI can help guide toward the type of cordectomy that is necessary, thus reducing the risk of both over- and undertreatment. Indeed, if epithelial hydrodissection is not complete after SI, the risk of finding an invasive SCC is 36%, according to our data, which justifies its excision by Type II cordectomy [5], achieving both safer oncologic outcomes [24] and a good vocal result [25]. On the other hand, the rate of false positives, due to an incomplete hydrodissection after SI (with histology negative for SCC), can be explained by the presence of adherences and scars between the epithelium and the vocal ligament (sulcus vocalis) [13,23].

Our results confirm the high predictive value of pathognomonic features obtained by coupling biologic endoscopy and VLS, as seen in Group A, in which just 1% of the excised lesions did not hide dysplastic foci beneath. The evidence of invasive or microinvasive SCC in 63% of such cases confirmed the lack of utility in performing SI when a pathologic vascular network is visible within a lesion or when an impairment of mucosal wave is seen at VLS. In such patients, in fact, Type II cordectomy should be always performed straightforwardly as an excisional biopsy, without previous tissue sampling. 

In our study, the higher adequacy of treatment (*p* = 0.006), the lower rate of deep positive margins after subligamental cordectomy (*p* = 0.03) in Group C, and the confirmatory results obtained by the multivariable analysis all highlight the importance of intraoperative SI in choosing the most adequate treatment for lesions amenable to a subepithelial cordectomy after proper endoscopic evaluation.

Although our results are obtained from a retrospective study containing a low level of evidence, these suggest that the routine use of such a comprehensive workup can guide the laryngologist to the correct surgical treatment, in terms of excisional biopsy within free margins, sparing the biologic and economic costs of a second surgical procedure, from both a vocal outcome and hospitalization point of view [19]. Further prospective studies, also analyzing in detail the costs and vocal and oncologic outcomes applying this kind of workup, are needed to further delineate the best treatment indications for such a clinical setting.

## 5. Conclusions

Excisional biopsy by Type I–II cordectomies after a comprehensive pre- and intraoperative diagnostic workup (such as that herein described), if confirmed by further prospective studies with a higher level of evidence, could be accepted as the most adequate treatment for unilateral glottic lesions not involving the anterior commissure and without impairment of vocal cord mobility or intraoperative evidence of vocal muscle involvement.

## Figures and Tables

**Figure 1 cancers-12-02165-f001:**
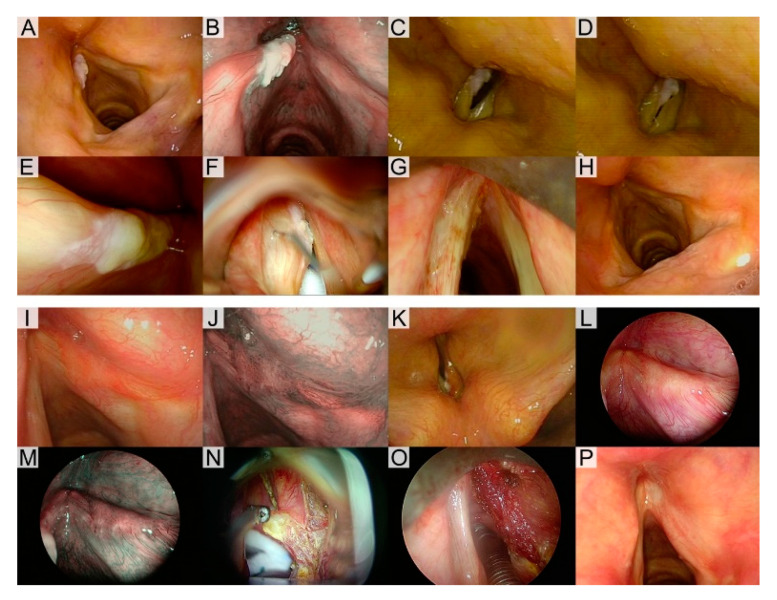
The application of the diagnostic workup. (**A**–**H**) Case 1. Preoperative white light (WL) videoendoscopy (**C**) showing a left mid-cord leukoplakia without narrow band imaging (NBI) vascular abnormalities (**B**). Mucosal wave was not impaired at videolaryngostroboscopy (VLS) (**C**,**D**). Intraoperative 70° endoscopy (**E**) confirmed the extension of the lesion to be confined to the left mid-cord. Complete ballooning was observed after saline infusion (SI) into the Reinke’s space (**F**) and, consequently, a subepithelial cordectomy (Type I) was performed (**G**). The histopathological diagnosis was squamous intraepithelial neoplasia (SIN2) and the patient is free of disease 18 months after surgery (**H**). (**I**–**P**) Case 2. Preoperative WL videoendoscopy (**I**) showing a right mid-cord erythroleukoplakia with NBI perpendicular vascular abnormalities (**J**). Mucosal wave was not impaired at VLS (**K**). Intraoperative 70° endoscopy (**L**–**M**) confirmed the extent of the lesion to be confined to the right mid-cord. The suspiciousness of the lesion guided the choice towards a subligamental cordectomy (Type II) (**N**–**O**). The final histopathological diagnosis was carcinoma in situ (CIS), resected in free margins, and the patient is now free of disease 33 months after surgery (**P**).

**Figure 2 cancers-12-02165-f002:**
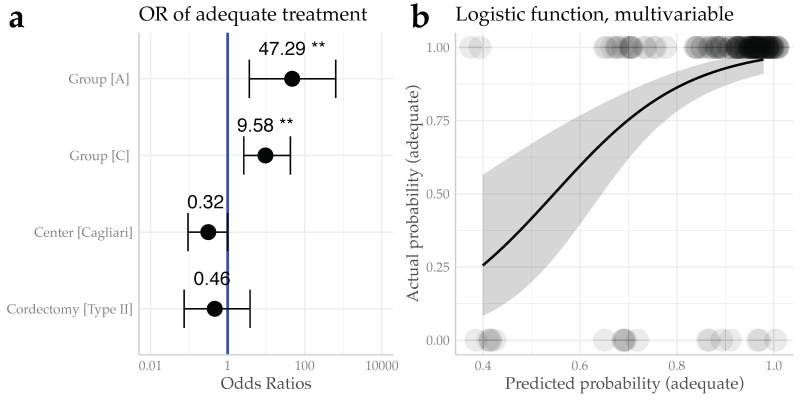
Multivariable logistic regression results. (**a**) Forest plot showing the Odds Ratios (ORs) for the probability of adequate treatment investigating the Group and Center variables. (**b**) Logistic function showing the predicted probability of adequacy of the treatment against the actual one. *Legend*: OR, Odds Ratio; reference levels for OR estimation: Group B, Genoa Center, and Type I cordectomy; ** *p* < 0.01.

**Table 1 cancers-12-02165-t001:** Definition of adequacy of treatment.

Histology	Treatment	
	Type I Cordectomy	Type II Cordectomy
Keratosis without atypia	Adequate	Not adequate
SIN1-2	Adequate	Adequate
SIN3-CIS	Adequate	Adequate
SCC	Not adequate	Adequate

*Legend:* SIN: squamous intraepithelial neoplasia (1, mild; 2, moderate; 3, severe); CIS: carcinoma in situ; SCC: squamous cell carcinoma.

**Table 2 cancers-12-02165-t002:** Histopathologic diagnosis and adequacy of treatment among groups.

Variables	All	Suspicious for Invasive SCC	Not Clearly Suspicious for Invasive SCC		
		Group A (BE + VLS)	Group B (BE + VLS)	Group C (BE + VLS + SI)	
		Type II Cordectomy	Type I Cordectomy	Type I Cordectomy	Type II Cordectomy
	N (%)	N (%)	N (%)	N (%)	N (%)
**Histology**					
Keratosis without atypia	26 (17)	***1 (1)***	4 (16)	19 (37)	***2 (18)***
SIN1-2	47 (30)	6 (9)	10 (40)	26 (51)	5 (45)
SIN3-CIS	21 (14)	18 (26)	1 (4)	2 (4)	0 (0)
SCC	61 (39)	43 (63)	***10 (40)***	***4 (8)***	4 (36)
**Adequacy of treatment**					
Adequate	138 (89)	67 (99)	15 (60)	47 (92)	9 (82)
Not adequate	17 (11)	1 (1)	10 (40)	4 (8)	2 (18)

*Legend:* SCC, squamous cell carcinoma; BE, biologic endoscopy; VLS, videolaryngostroboscopy; SIN, squamous intraepithelial neoplasia; CIS, carcinoma in situ; bold underlined italic formatting for not adequate treatments.

**Table 3 cancers-12-02165-t003:** Association between histopathologic diagnosis and endoscopic appearance.

Histopathologic Diagnosis	Endoscopic Findings		
	Suspicious for SCC	Not clearly Suspicious for SCC	*p*
	N (%)	N (%)	
**Keratosis without atypia**	1 (1)	25 (29)	<0.001
**SIN 1-2**	6 (9)	41 (47)	
**SIN3-CIS**	18 (26)	3 (3)	
**SCC**	43 (63)	18 (21)	

*Legend:* SCC, squamous cell carcinoma; SIN, squamous intraepithelial neoplasia; CIS, carcinoma in situ; *p* value estimated by the Chi-square test.

**Table 4 cancers-12-02165-t004:** Association between Groups B and C and adequacy of treatment.

Not Clearly Suspicious for Invasive SCC		All	Treatment		
			Adequate	Not Adequate	
		N (%)	N (%)	N (%)	*p*
**Workup**	**Group B (BE + VLS)**	25 (29)	15 (60)	10 (40)	0.006
	**Group C (BE + VLS + SI)**	62 (71)	56 (90)	6 (10)	

*Legend:* SCC, squamous cell carcinoma; BE, biologic endoscopy; VLS, videolaryngostroboscopy; SI, saline infusion into the Reinke space (*p* value estimated by the Fisher’s exact test).

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
