# Peer review of "Laryngeal Mid-Cord Erythroleukoplakias: How to Modulate the Transoral CO2 Laser Excisional Biopsy"

_cancers, 2020, doi:10.3390/cancers12082165_

Round 1

Reviewer 1 Report

Dear authors,

I thank you for your manuscript. Please find my comments below:

  1. Consider to refer article : Evaluation of vascular patterns using Contact Endoscopy and Narrow Band Imaging (CE-NBI) for the diagnosis of vocal fold malignancy. Nikolaos Davaris et. al. MDPI January 2020
  2. Please explain in Materials and methods how you define a treatment as “adequate” or not.
  3. Line 194: twice after.
  4. Line 214: In conclusion: please note about the retrospective character of your study and its limitation.
  5. Line 216: if possible choose other accent for your words choice such as “ should be accepted as the most adequate treatment”. Because of low degree of evidence of “retrospective studies”.

Author Response

Reviewer 1

  1. Consider to refer article : Evaluation of vascular patterns using Contact Endoscopy and Narrow Band Imaging (CE-NBI) for the diagnosis of vocal fold malignancy. Nikolaos Davaris et. al. MDPI January 2020

Response (1) Thanks for this suggestion, we added in Discussion section the following comment including PMID31968528 and PMID25582112 references: Lines 349-51 “Furthermore, coupling biologic endoscopy with contact endoscopes is a promising and interrater reliable technique to further improve the diagnostic accuracy of intraoperative endoscopic evaluation [20,21]..”

  1. Please explain in Materials and methods how you define a treatment as “adequate” or not.

Response (2) In methods section we better stated “2.4 Outcomes evaluation and statistical analysis

The adequacy of treatment, in terms of type of cordectomy chosen to minimize the damage of the multilayered vocal cord anatomy according to the findings of the above-mentioned diagnostic workup, was calculated in relation to the final histopathological report (Table 1).” Lines 226-29

In Page 5 lines 232-38 full details of this explanation were given with appropriate references (PMID: 12783241, PMID: 10892699, PMID: 11558755) and Table 1 was built to this aim.

  1. Line 194: twice after.

Response (3) Thanks, we corrected it.

  1. Line 214: In conclusion: please note about the retrospective character of your study and its limitation.
  2. Line 216: if possible choose other accent for your words choice such as “ should be accepted as the most adequate treatment”. Because of low degree of evidence of “retrospective studies”.

Response (4,5) Thanks for these suggestion, we rephrased the conclusion paragraph underlying the need for new prospective studies with higher level of evidence, as following: “Excisional biopsy by Type I–II cordectomies after a comprehensive pre- and intraoperative diagnostic workup such as that herein described, if confirmed by further prospective studies with higher level of evidence, could be accepted as the most adequate treatment for unilateral glottic lesions not involving the anterior commissure, without impairment of vocal cord mobility or intraoperative evidence of vocal muscle involvement.”

Reviewer 2 Report

Laryngeal mid-cord erythroleukoplakias: How to 2 modulate the transoral CO2 laser excisional biopsy

The authors aimed to investigate a comprehensive workup for one-step diagnosis and treatment of mid-cord erythroleukoplakias with CO2 laser-assisted excisional biopsy.

The subject is of increasing interest.

It is not clear whether the list of authors is complete or not; after the conjunction "and" no surname is indicated.

The affiliations of the authors and the contact details of the corresponding author are missing.

INTRODUCTION. Quite well written section. Although in a different context, the limits of minute glottic biopsies in differentiating malignant and benign lesions have been recently emphasized (DOI: 10.1136/jclinpath-2015-203308). Primary and secondary end-points should be defined and described more in detail. The logic of the distinction in three groups (A, B, C) and therefore their essential characteristics should be briefly introduced.

METHODS. Groups B and C features should be better reported. CO2 laser–assisted excision technique should be described more in detail. The statistical approach is very simple but overall appropriate. Statistical methods: please change “Fischer’s exact test” with “Fisher's exact test”.

RESULTS. To increase the clarity of the section it is appropriate to be divided into subsections. Given the importance of the repeatability of the diagnostic approach, were there significant differences in the results of the two centers involved?

DISCUSSION. A para should be devoted to critically analyze more in depth strengths and weaknesses of the present study. A paragraph should also be dedicated to commenting in more detail the potential role in diagnostic, therapeutic and prognostic terms of the preliminary evidence obtained.

The section order should be as follows: 1. Introduction; 2. Results 3. Discussion; 4. Materials and Methods; 5. Conclusion.

Author Response

Reviewer 2

Laryngeal mid-cord erythroleukoplakias: How to 2 modulate the transoral CO2 laser excisional biopsy

The authors aimed to investigate a comprehensive workup for one-step diagnosis and treatment of mid-cord erythroleukoplakias with CO2 laser-assisted excisional biopsy.

The subject is of increasing interest.

It is not clear whether the list of authors is complete or not; after the conjunction "and" no surname is indicated.

The affiliations of the authors and the contact details of the corresponding author are missing.

Response: Unfortunately, there has been some mistake after the file uploading for the first submission, now the text is complete again with full affiliations. The list of authors was correct.

INTRODUCTION. Quite well written section. Although in a different context, the limits of minute glottic biopsies in differentiating malignant and benign lesions have been recently emphasized (DOI: 10.1136/jclinpath-2015-203308). Primary and secondary end-points should be defined and described more in detail. The logic of the distinction in three groups (A, B, C) and therefore their essential characteristics should be briefly introduced.

Response:  We extended the last paragraph of the introduction to better guide directly to the Results (having moved the M&M section after Discussion). Here we also introduced the differences among groups of treatment.

Lines 87-99 “…The pre- and intraoperative workup included preoperative flexible videoendoscopy and VLS, and intraoperative endoscopy with WL and biologic endoscopy (NBI or SPIES) assessment. According to the information gained, patients were subsequently non-randomly divided into 3 groups of treatment. Group A was composed of patients harboring highly suspicious lesions and straightforward treated by a subligamental cordectomy (Type II). Groups B and C were composed of patients presenting non-clearly suspicious lesions, in which the data emerging from the diagnostic workup were not suggestive for SCC. In Group B, upfront excisional biopsy by a subepithelial (Type I) cordectomy was performed, whereas in Group C a further investigation with intraoperative SI guided the final choice of performing a subepithelial (Type I) or subligamental (Type II) cordectomy [12]. The main outcome was the adequacy of the treatment, while the second one was represented by the investigation of the potential value of SI testing in non-clearly suspicious lesions.

METHODS. Groups B and C features should be better reported. CO2 laser–assisted excision technique should be described more in detail. The statistical approach is very simple but overall appropriate. Statistical methods: please change “Fischer’s exact test” with “Fisher's exact test”.

Response:  We better detailed the features of the groups and the technique used for excisional biopsy performance as following:

Lines 207-224: “2.3 Treatments

Surgical procedures encompassed CO2 TOLMS subepithelial (Type I) or subligamental (Type II) cordectomies, as defined according to the ELS nomenclature [13]. Patients were intubated with a laser-safe endotracheal tube 5.0-6.0 mm in internal diameter (Shiley™ Laser Oral Endotracheal Tubes, Medtronic Xomed, Jacksonville, FL, USA) under general anesthesia. The laryngeal exposure was achieved by large-bore laryngoscopes (Microfrance 121; Microfrance, Paris, France) or small-bore ones for unfavorable exposures (Dedo-Ossof; Pilling, Philadelphia, PA). The Boston University suspension system (Pilling; Philadelphia, PA) was employed. A UltraPulse® DUO CO2 Laser coupled to a Digital AcuBlade scanning micromanipulator (Lumenis, Yokneam, Israel), set on continuous ultrapulse mode (line shape 0.9-1.2 mm), at 2.5-3.5 watts of delivered power at 400 mm of focal length was used as the cutting device.

The subepithelial (Type I) cordectomy was performed resecting the vocal fold epithelium with the deep plane passing through the superficial layer of the lamina propria and thus sparing the deeper layers and vocal ligament. Subligamental (Type II) cordectomy included the resection of the epithelium, Reinke’s space and vocal ligament, having as the deep plane of resection the spared vocal muscle. In order to expose the entire vocal fold, partial resection of the ventricular band may be necessary. For both types of cordectomy the specimen is en-bloc resected and oriented for proper histopathologic assessment of surgical margins.”

We apologize for the mistake and corrected “Fischer’s exact test” with “Fisher's exact test”.

RESULTS. To increase the clarity of the section it is appropriate to be divided into subsections. Given the importance of the repeatability of the diagnostic approach, were there significant differences in the results of the two centers involved?

Response:  We divided the Results section into: 3.1 Patient cohort; 3.2 Univariable adequacy of treatment analysis; 3.3 Multivariable adequacy of treatment analysis; 3.4 Margins control. The same was done also for M&M section improving the clarity of this section too.

Furthermore, we improved the analysis including a multivariable logistic regression model investigating as covariates of interest the Groups, Center and Cordectomy performed for the prediction of the Adequacy of treatment as dependent variable, confirming the results obtained by the univariable analysis: Group A and C are significantly associated with higher probability of adequate treatment, compared to Group B (Type I cordectomy without SI assessment), weighted for the other covariates; no significantly differences were observed among the two Centers or Cordectomy performed.

Lines 293-306 and Figure 2: “3.3 Multivariable adequacy of treatment analysis

Investigating by a multivariable logistic regression model the Group of treatment, the Center in whom surgery was performed, and the cordectomy applied as possible confounders, Group A [OR 47.29 CI95% (3.66-637.88), p=0.002) and Group C [OR 9.58 CI95% (2.63-42.52), p=0.001] were confirmed to be associated with a significantly higher probability of adequacy of treatment, compared to Group B. No significant difference was observed for Cagliari center compared to Genoa one [OR 0.32 CI95% (0.09-1.00), p=0.052], and for Type II compared to Type I cordectomy [OR 0.46 CI95% (0.07-3.82), p=0.42], as shown in Figure 2A-B.

Figure 2: Multivariable logistic regression results. (a) Forest plot showing the Odds Ratios (OR) for the risk of adequate treatment investigating the Group and Center variables. (b) Logistic function showing the predicted probability of non-adequacy of the treatment against the actual one.

Legend: OR, Odds Ratio; References levels for OR estimation: Group B, Genoa Center and Type I cordectomy); significance levels: *p<0.05, **p<0.01; ***p<0.001.

DISCUSSION. A para should be devoted to critically analyze more in depth strengths and weaknesses of the present study. A paragraph should also be dedicated to commenting in more detail the potential role in diagnostic, therapeutic and prognostic terms of the preliminary evidence obtained.

Response:  The last two paragraph of the discussion were expanded underlying the limitation of the study and suggestions for future perspectives: “In our study, the higher adequacy of treatment (p=0.006), the lower rate of deep positive margins after subligamental cordectomy (p=0.03) in Group C, and the confirmatory results obtained by the multivariable analysis all highlight the importance of intraoperative SI in choosing the most adequate treatment for lesions amenable to a subepithelial cordectomy after proper endoscopic evaluation.

Although our results are obtained from a retrospective study having a low level of evidence, these suggest that the routine use of such a comprehensive workup can guide the laryngologist to the correct surgical treatment in terms of excisional biopsy within free margins, sparing the biologic and economic costs of a second surgical procedure, from both the vocal outcome and hospitalization points of view.[18] Further prospective studies, analyzing in details also the costs, vocal and oncologic outcomes applying this kind of workup are needed to furtherly delineate the best treatment indications for such a clinical setting.”

The section order should be as follows: 1. Introduction; 2. Results 3. Discussion; 4. Materials and Methods; 5. Conclusion.

Response: For a better clarity of the paper, if approved by the Editor, we would rather to keep the conventional order of: 1. Introduction; 2. Materials and Methods 3. Results; 4. Discussion; 5. Conclusion. Other articles published in Cancers were approved with this kind of order of the sections.

Round 2

Reviewer 2 Report

Laryngeal mid-cord erythroleukoplakias: How to 2 modulate the transoral CO2 laser excisional biopsy

The paper has been reasonably improved by authors revisions according to Reviewers’ suggestions.

I would like to point out only a few minor issues:

INTRODUCTION. It was suggested the to mention the limits of minute glottic biopsies in differentiating malignant and benign lesions also considering DOI: 10.1136/jclinpath-2015-203308).

There are some typos in References. See: Ref. 1 “L.U.C.A.” and “P.I.E.R.O.”.

Author Response

Thanks for the suggestion, we improved the Introduction section adding the suggested reference: " To further complicate the situation, multiple biopsies (with or without frozen section analysis) are frequently inadequate in obtaining a precise diagnosis and subsequently in guiding the ensuing treatment of glottic leuko- or erythroleukoplakias, since they are not representative of the entire lesion and less suitable for further biomarkers analysis, if needed, to provide an accurate differential diagnosis [2]"

We corrected the typos in References

.